# Removal of Cobalt (II) from Waters Contaminated by the Biomass of *Eichhornia crassipes*

**Ismael Acosta-Rodríguez [1], Adriana Rodríguez-Pérez [2], Nancy Cecilia Pacheco-Castillo [1], Erika Enríquez-Domínguez [1], Juan Fernando Cárdenas-González [2,*] and Víctor-Manuel Martínez-Juárez [3]**

[1] Laboratorio de Micología Experimental, Facultad de Ciencias Químicas, Universidad Autónoma de San Luís Potosí, 78320 San Luís Potosí, Mexico; iacosta@uaslp.mx (I.A.-R.); nancy.cecilia.pc@gmail.com (N.C.P.-C.); erika.dominguez@uaslp.mx (E.E.-D.)

[2] Unidad Académica Multidisciplinaria Zona Media, Universidad Autónoma de San Luís Potosí, 79617 Ríoverde, Mexico; sarai.rodriguez@uaslp.mx

[3] Area Académica de Medicina Veterinaria y Zootecnia, Universidad Autónoma del Estado de Hidalgo, 43600 Hidalgo, Mexico; victormj@uaeh.edu.mx

* Correspondence: juan.cardenas@uaslp.mx

**Abstract:** Due to the increase in contamination of aquatic niches by different heavy metals, different technologies have been studied to eliminate these pollutants from contaminated aquatic sources. So the objective of this work was to determine the removal of cobalt (II) in aqueous solution by the biomass of the aquatic lily or water hyacinth (*Eichhornia crassipes*) which, is one of the main weeds present in fresh water, due to its rapid reproduction, growth, and high competitiveness, by the colorimetric method of the methyl isobutyl ketone. The removal was evaluated at different pHs (4.0–8.0) for 28 h. The effect of temperature in the range from 20 °C to 50 °C and the removal at different initial concentrations of cobalt (II) of 100 to 500 mg/L was also studied. The highest bioadsorption (100 mg/L) was at 28 h, at pH 5.0 and 28 °C, with a removal capacity of 73.1%, which is like some reports in the literature. Regarding the temperature, the highest removal was at 50 °C, at 28 h, with a removal of 89%. At the metal and biomass concentrations analyzed, its removal was 82% with 400–500 mg/L, and 100% with 5 g of natural biomass at 20 h. In addition, this completely removes the metal in situ (100 mg/L in contaminated water, at 7 days of incubation, with 10 g of natural biomass in 100 mL). So, the natural biomass can be used to remove it from industrial wastewater，even if in vivo, only eliminate 17.3% in 4 weeks.

**Keywords:** contaminants; heavy metals; removal; cobalt; water hyacinth; biomass

## 1. Introduction

The great industrial growth has produced a progressive increase in wastewater discharges from the same and, therefore, a deterioration in water quality. Pollutants pose a danger to both human and environmental health. Some pollutants are of organic origin, such as hydrocarbons and pesticides, and inorganic, such as heavy metals and dye, which play a fundamental role due to their importance and potential danger [1,2]. Although heavy metals are present in water naturally, these are usually found at levels less than 1 mg/L, despite their low concentration levels, they have important chemical and biological implications in systems natural aquatic [3]. When they are above certain concentrations, as occurs in the case of some discharges, they become harmful both for the environment and for living beings, being able to affect the normal metabolism of an organism, either because they bind to non-specific biomolecules or because they cause oxidative damage due to its ability to catalyze oxidation-reduction reactions. This can lead to a deactivation of essential enzymatic reactions, damage to cell structures or DNA (mutagenesis). In humans, short exposures to high levels of certain metals can cause symptoms of acute toxicity, while long exposures to lower levels can lead to allergies or even cancer [4]. Some metals that

are of great toxicological and exotoxicological importance are: mercury, chromium, lead, cadmium, nickel, cobalt, and zinc, which, once released into the environment, accumulate and concentrate in the soil and sediments, where they can remain for hundreds of years affecting ecosystems. Therefore, it is more feasible to control the problem from the source of emission before they reach the environment [5]. Currently, the most used methods to remove heavy metals in wastewater are chemical precipitation, filtration, coagulation, solvent extraction, electrolysis, membrane separation, ion exchange, and sorption techniques [6]. Some of these processes can become expensive, given the high operating costs and requirements energetic, as is the case of polymeric resins, which are quite efficient, but not very economical. For example, although ion exchange useful for effective water purification, resins are expensive due to that its synthesis is complex and fossil resources are used as the basis for its preparation. On the other hand, electrochemical treatments, and methods of membrane separation systems are relatively expensive, either because they require sophisticated equipment and because of the high energy expenditure associated with their functioning [6]. The process known as sorption refers to the capacity of certain materials to bind to anions and cations (as is the case with metals). During the sorption, metals bind to active centers found on the surface of materials through different mechanisms [1], and these generally include physical adsorption, ion exchange adsorption, chelation, complex formation, microprecipitation, visible-light-responsive photocatalysts [7], or absorption [8]. The sorption with biological sorbents is also known as biosorption and, as indicated, it can be used in remediation processes with low cost and high efficiency, allowing the minimization of chemical waste or biological [9]. It happens when metal cations are joined by electrostatic interactions to the anionic sites found in biosorbents. These sites, that serve as active centers for biosorption, are in the carboxyl, hydroxyl, amino, or sulfonic groups, which are part of the structure of most naturally occurring polymers [6,10]. In Mexico, agribusiness is one of the most important activities due to its growth in recent years, and it is the one that generates the most by-products that are not used [11], among which are: coffee bagasse, agave, maguey, sugar cane, straws from different crops, organic residues of fruits and vegetables [12]. In this regard, the use of different plant products and other materials like nanocomposite, glutathione-magnetite [13], and natural locust bean gum-based hydrogels [14], with the ability to accumulate and/or bioadsorb heavy metals and other pollutants has been reported.

On the other hand, the water lily or water hyacinth (*Eichhornia crassipes*) is an aquatic macrophyte plant, which is used as an ornamental species for its showy flowers in ponds and aquariums, it is native from South America mainly in the Amazon and Silver basins, it came to Mexico at the end of the 19th century, where it spread rapidly until it became a plague spreading throughout the country, which is considered one of the worst weeds in the world [15]. The only places where it has not been recorded are the states of Baja California Sur, Tlaxcala, and Zacatecas [16]. The water lily has been reported almost always alive, to accumulate and remove different pollutants, among which are heavy metals, for example: the accumulation of copper and cobalt [17], the decrease of arsenic from waters of a town in Peru [18], the accumulation of silver [19], removal of cadmium from wastewater [20], phytoremediation of cyanided water [21], the phytoextraction of arsenic, cadmium, and copper in an artificial wetland with different plants [22], the removal of different heavy metals by the biomass of different aquatic plants (*E. crassipes, Potamogeton lucens* and *Salvinia herzegoi*) [23,24], the phytoremediation of an effluent contaminated with mercury in a wetland in Antioquia (Colombia) [25], the elimination of mercury, cadmium, and arsenic from wastewater [26], and the phytoremediation of methylene blue by the same plant [27]. Therefore, the objective of this work was to evaluate the adsorbent capacity of the biomass of the water lily (*E. crassipes*) in the removal of cobalt (II) in an aqueous solution.

## 2. Materials and Methods

### 2.1. Bioadsorbent Used

Initially, the water lily plant was obtained from the San José Dam, San Luis Potosí, S.L.P. Mexico (Figure 1a), located to the west of the city, it is the best known dam in San Luis Potosi, it has an area of 344 hectares, inaugurated on 3 September 1903, with the capacity to hold up to 10 million cubic meters of water retained by a curtain of 32 m high and 100 m long, built almost entirely from a quarry. It is the most prominent dam in the state, and it is a protected natural area in the form of an urban park. To obtain the biomass, the plant was washed 24 h with EDTA at 10% (p/v), and later 1 week in tridesionized water with constant agitation, with water changes every 12 h, and boiled for 60 min to remove dust and adhered organic components, and washed again under the same conditions (Figure 1b). It was dried at 80 °C for 24 h in a bacteriological oven, ground in a blender, and stored in amber flasks until use (Figure 1c).

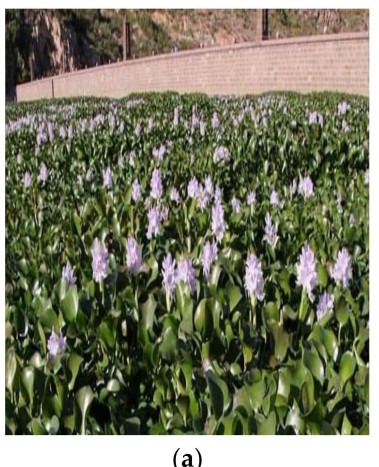
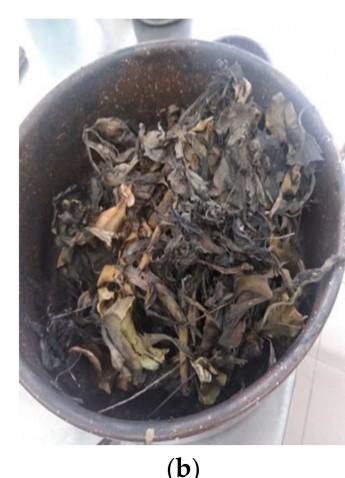
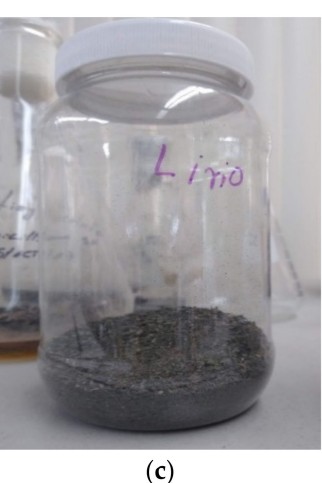

| (a) | (b) | (c) |

**Figure 1.** Natural biomass of water lily utilized: (**a**) water lily growth in San José Dam. San Luis Potosí, S.L.P. México (**b**) Water lily collected and washed (**c**) Stored and ground biomass.

### 2.2. Cobalt Solutions

We worked with 100 mL of a 100 mg/L solution of cobalt (II) obtained by diluting 1.0 g/L of standard solution prepared in tridesionized water from $CoCl_2$ (Analit Brand, St. Petersburg, Russia). The pH of the dilution to be analyzed was adjusted to the desired value (4, 5, 6, 7, and 8), with 1 M $H_2SO_4$ and/or 1 M NaOH, before adding it to the biomass.

### 2.3. Determination of the Concentration of Divalent Cobalt

Cobalt (II) concentration in aqueous solution was determined by UV spectroscopy in a double-beam UV-Visible spectrophotometer Shimadzu UV-2101PC, using methyl isobutyl ketone [28]. In a separation funnel, to 500 μL of the test solution (100 mg/L of cobalt II), add 0.5 mL of 20%/(p/v) ammonium thiocyanate solution, then add 5 mg of fluoride sodium (to eliminate contaminating ions in the sample), finally, 4 mL of methyl isobutyl ketone are added, stirred gently for 1 min and left to rest for 10 min observing the development of a blue color, and the absorbance of the sample is read at 622 nm. Samples were taken at different times, and the natural biomass was removed by centrifugation (3000 rpm/5 min), and the supernatant was analyzed to determine the concentration of the metal in solution. The instrument was calibrated to zero with a blank of the reagents used without the addition of the metal, determining the cobalt concentration with a calibration curve of 0–100 mg/L of the metal. All experiments were performed 3 times and in duplicate.

## 3. Results

### 3.1. Effect of Incubation Time and pH

Initially, the bioadsorption of 100 mg/L of cobalt (II) was analyzed, at different incubation times at the following pH values: 4.0, 5.0, 6.0, 7.0, and 8.0 finding that at pH 5.0, 73.1% of the metal was removed at 28 h (Figure 2).

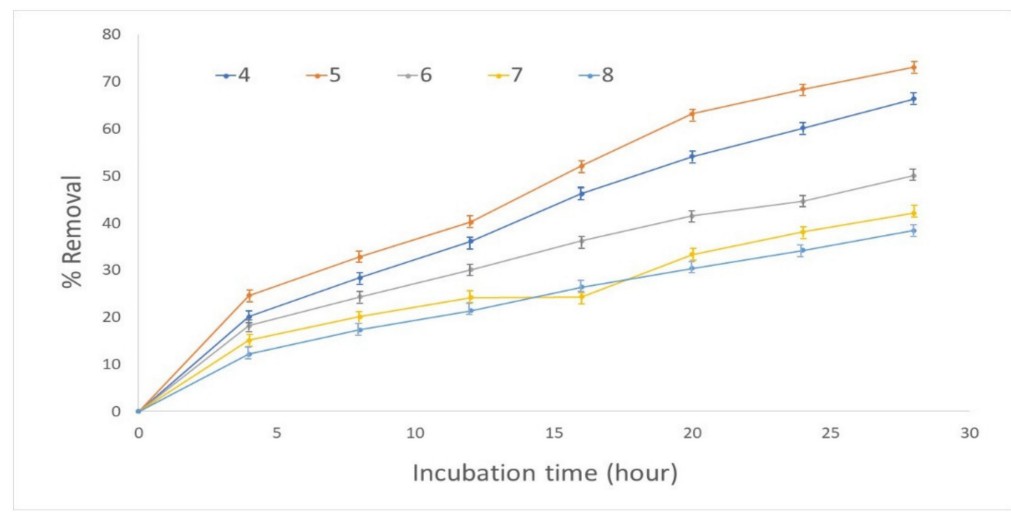

**Figure 2.** Effect of pH and incubation time on the removal of cobalt (II) by *E. crassipes*. 100 mg/L cobalt (II). 28 °C. 100 rpm. 1 g of natural biomass.

### 3.2. Effect of Temperature

In relation to the temperature, the highest removal was observed at 50 °C, since at 28 h 88.9% of the metal in solution is removed, while at 20 °C, 58.3% is removed at the same time (Figure 3).

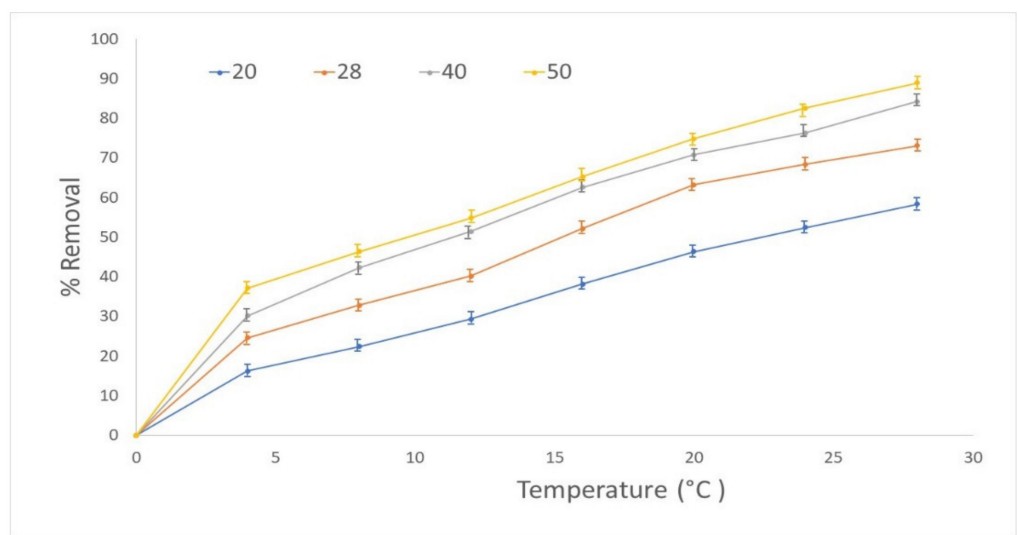

**Figure 3.** Effect of temperature on the removal of cobalt (II) by *E. crassipes*. 100 mg/L cobalt (II). pH 5.0. 100 rpm. 1 g of natural biomass. 28 h of incubation.

### 3.3. Effect of the Concentration of Cobalt (II)

Regarding the effect of different concentrations of cobalt (II) in solution, on its removal, at a pH of 5.0 ± 0.2, with 1 g of *E. crassipes* biomass, at 28 °C, and 100 rpm, it was found that, at a higher concentration of the metal, the removal is greater, up to 400–500 mg/L of the metal, it removal 82% of it at 28 h of incubation (Figure 4).

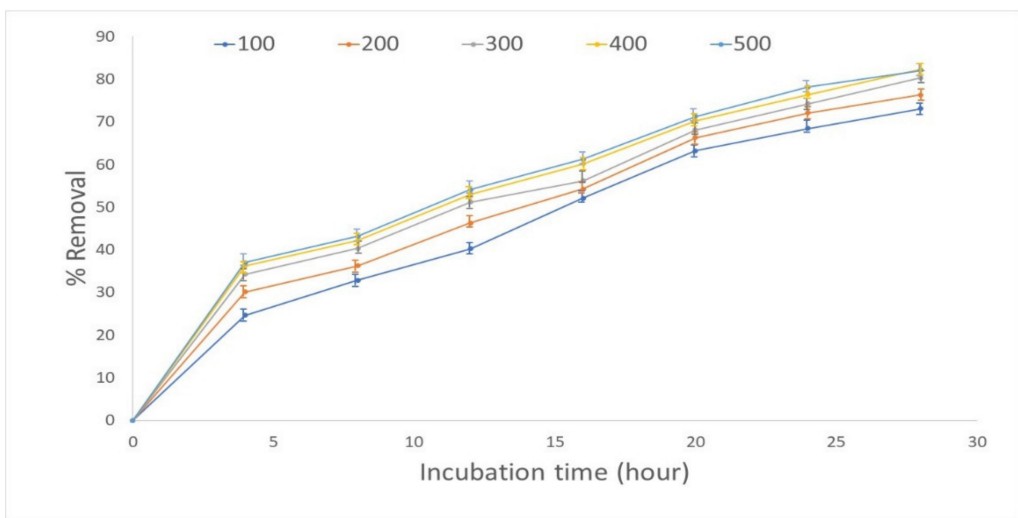

**Figure 4.** Effect of concentration of cobalt (II) (mg/L) on the removal of cobalt (II) by *E. crassipes*. pH 5.0. 28 °C. 100 rpm. 1 g of natural biomass. 28 h of incubation.

### 3.4. Effect of Initial Biomass Concentration

In Figure 5, the effect of biomass concentration on metal removal is observed. If the concentration of this is increased, the elimination of the metal in solution is increased, because with 5 g of biomass, 100% is removed after 20 h, while with 1 g of biomass, 63.2% is removed in the same incubation time.

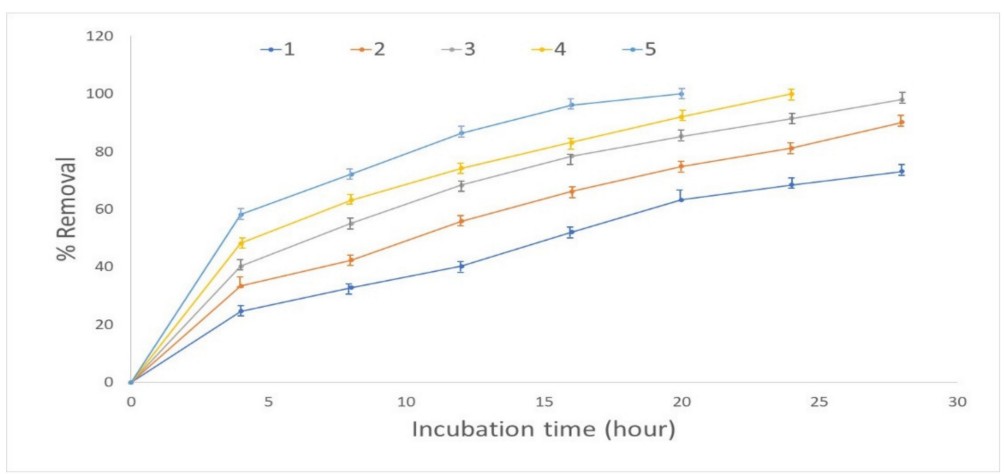

**Figure 5.** Effect of *E. crassipes* biomass (g/L) on the removal of 100 cobalt (II). pH 5.0. 28 °C. 100 rpm.

### 3.5. Removal of Co (II) in Industrial Wastes

A cobalt (II) bioremediation test was carried out from water contaminated with 100 mg/L of the metal (adjusted), obtained from an industrial effluent lagoon (Tenorio Tank) located to the east of the capital city of San Luis Potosí, S.L.P. Mexico, with 5 and 10 g of natural biomass, observing that this removal 90.1% and 100% of the metal, respectively, at 7 days of incubation (Figure 6).

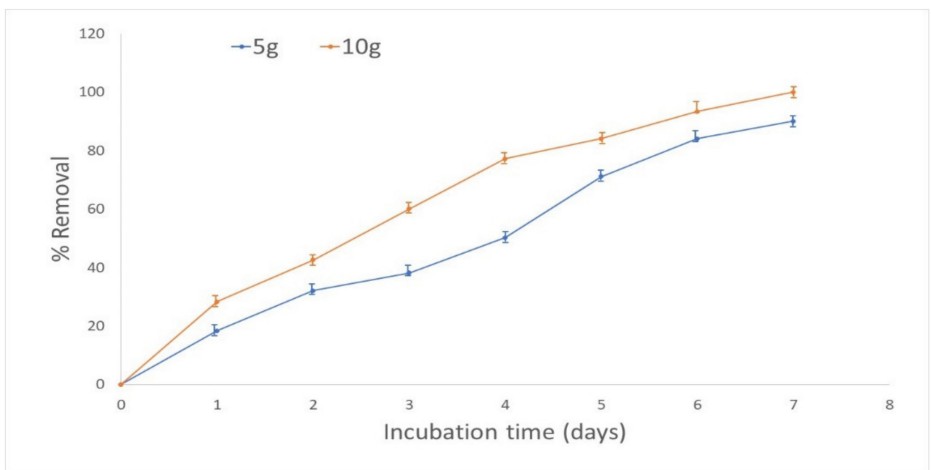

**Figure 6.** Cobalt (II) bioremediation by *E. crassipes* biomass from contaminated water (100 mL) with 100 mg of cobalt (II)/L. pH 5.0 (adjusted). 28 °C. 100 rpm. 5 and 10 g of biomass.

### 3.6. Removal of Cobalt (II) In Vivo

Finally, an in vivo experiment was carried out with water lily plants (lily A and lily B), incubating them in the presence and absence of a 100 mg/L solution of cobalt (II) (adjusted) and pH 6.8, in a final volume of 250 mL, from a lagoon of industrial effluents, taking 5 mL samples every week for a month, determining the concentration of the metal in the supernatant, finding that at 4 weeks of incubation, 17.3% of the metal under study was removed (Figure 7), which is less than that reported in the previous experiment, and also, the plant that was incubated in the presence of the metal, grew less and slowly lost its green color (data not shown).

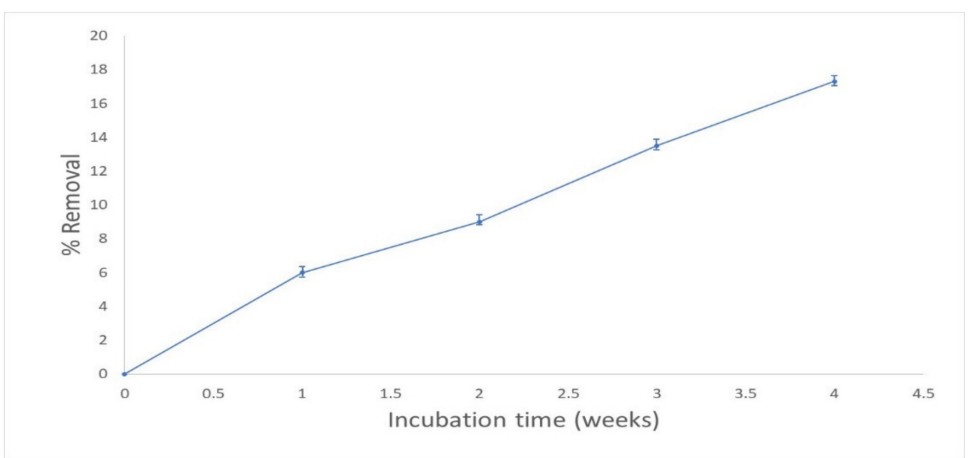

**Figure 7.** In vivo removal of 100 mg/L cobalt (II) (adjusted) from water contaminated (250 mL) by *E. crassipes* live. pH 6.8. 28 °C. Static conditions.

## 4. Discussion

Cobalt has an irreplaceable functionality in modern technology, it is an element that occurs naturally in the earth's crust. It is a very small part of our environment. It is a component of vitamin B12, which helps in the production of red blood cells. Animals and humans need very small amounts to stay healthy. Metal poisoning can occur when the human is exposed to large amounts of this element, an excessive metal exposure can result in a range of symptoms/conditions in humans including goitre and reduced thyroid activity [29]. There are three basic ways that cobalt can cause poisoning: ingest it in excess, inhale it in large quantities into the lungs, or by constant contact with the skin. EPA has not established a Reference Concentration (RfC) or a Reference Dose (RfD) for cobalt.

The California Environmental Protection Agency 3 (CalEPA) has established a chronic reference exposure level of 0.000005 milligrams per cubic meter (mg/m$^3$) for cobalt based on respiratory effects in rats and mice [30,31]. Therefore, it is very important to try to eliminate it from the different environmental niches, using technologies like natural biomasses, that are efficient, cheap, and that do not pollute the environment anymore [32,33].

In this work, we analyze the application of the water lily plant biomass for the removal of cobalt (II) in solution. Initially, the bioadsorption of the metal was analyzed, at different incubation times and pH values finding that at 28 h and pH 5.0, the removal was 73.1%. In this regard, the literature reports an optimal incubation time of 24 h with removal of the same metal of 93%, 77.5%, and 70.4%, for the fungus *Paecilomyces* sp., *Penicillium* sp. and *Aspergillus niger*, respectively [33], 20 min with 20 mg/L using *Citrus lemon* leaves powder [34], 180 and 120 min for the removal of 25–27 mg/L (76%), and 24.3 mg/L (24.3%), with red marine algae *Callithamnion corymbosum* biomass [35], and XAD-4 resin modified with *Anoxybacillus kestanbolienis* [36], a time of 60 min for the removal of 2.557 mg/g, with *Luffa cylindrica* as biosorbent [37], 120 min for elimination of 57.34% of 100 mg/L with palm kernel mesocarp fiber modified [38].

With respect to pH, the optimum was 5.0 (73.1% of removal at 28 h) and has been reported a pH optimum of 2.0, 4.4, 5.0, 6.0, and 4.5 for *C. lemon* leaves powder and *C. corymbosum* biomass [34,35], XAD-4 resin modified with *A. kestanbolienis* [35], *L. cylindrica*, [37], and palm kernel mesocarp fiber modified [38] as biosorbents, respectively. This phenomenon can be explained based on the less competition between positively charged H$^+$ and Co$^{+2}$ ions for the similar functional group. As the pH rises, more ligands are exposed and the number of negatively charged groups on the adsorbent matrix probably increases, improving the removal of the cationic species [39]. Also, it was found that the higher temperature, the bioadsorption of the metal is greater. In this regard, our results are similar for three fungal species the literature reports an optimal incubation time of 24 h with a maximum adsorption of 100%, 97.1%, and 94.1%, respectively, at 50 °C for *Paecilomyces* sp., *Penicillium* sp. and *A. niger*, respectively [33], for *P. catenlannulatus,* as the uptake of cobalt (II) increase with increasing temperature from 20–40 °C [40], *Cocos nucifera* L. leaf powder [41], and *F. benghalensis* leaf powder [42], but are different for *C. corymbosum* biomass, bark of Eucalyptus, and spent green tea leaves, for the removal of the same heavy metal, which the optimum temperature was 22 °C [9,35,43], and for the removal of cobalt (II) with *L. cylindrica* as biosorbent, where a range of temperature of 10 °C to 40 °C, not affect the removal [37]. On the other hand, enhancement of the adsorption capacity of the fungal biomasses, at higher temperatures may be attributed to the activation of the adsorbing surface, the accelerated diffusivity of metal with the increasing temperature and the increase in the mobility of metal ions [39].

Regarding the effect of different metal concentrations in solution, it was found that, at a higher concentration of this, the removal is greater. This may be due to the increased number of competing for the functional groups of the surface of the biomass ions [23], and this is similar for bark of Eucalyptus, with a range of 20–200 mg/L of metal concentration [43], for the Cyanobacterium *Spirulina platensis*, because the cobalt (II) biosorption, increased proportionately with increasing its concentration, attaining a maximum value of 181 mg cobalt (II)/g with 600 mg/L initial concentration of the metal [44], for leaves of *Tectona grandis* (teak) tree were collected from the farm lands in Vellore District, India [45], and for the fungi *Paecilomyces* sp., *Penicillium* sp. and *A. niger*, respectively, in which if increase the metal concentration of 50–300 mg/L, increase the percentage of removal [33]. However, it is different for the algae *H. valentiae*, for which a higher initial concentration of cobalt (II), decreases the percentage removal [39], too for the calcium alginate of seaweed *Macrocistis piryfera*, [46], rice straw natural and activated rice straw [47].

On the other hand, if the biomass concentration is increased, the elimination of the metal in solution is increased, with more biosorption sites of the same, because the amount of added biosorbent determines the number of binding sites available for metal biosorption. These results are similar for *C. lemon* leaves powder, in which the adsorption capacity of

cobalt (II) was increased with an increase in an adsorbent dose up to 0.1–0.5 g, however, the percent removal decreases at higher concentrations, and the maximum adsorption occurs at 60 ppm [34], as well as with 50 to 200 mg of XAD-4 resin modified with *A. kestanbolienis* [36], with 0.5–1.0 g for palm kernel mesocarp fiber modified [38], bark of Eucalyptus, with a range of 0.02–0.6 g/L of biomass concentration [43], and spent green tea leaves, with 0.04–0.25 mg/L of biomass [9], but are different for *C. corymbosum* biomass, in which the removal is not increased if the biomass concentration increases [35], and for *L. cylindrica* and palm kernel mesocarp fiber modified as biosorbent, in which if increase the biomass concentration of 10–100 mg/L, diminish the removal capacity [37,38].

It has been suggested that the water lily or water hyacinth has some advantages to treat industrial and domestic wastewater, sewage effluents, and sludge ponds, due to its high absorption capacity for different organic and inorganic pollutants, it can tolerate an extremely polluted environment and has a great capacity for biomass production [23,24]. Therefore, a cobalt (II) bioremediation test was carried out from water contaminated from an industrial effluent lagoon (Tenorio Tank), observing that this removal 90.1% and 100% of the metal, with 5 and 10 g of biomass, respectively at 7 days of incubation. which coincides with some reports in the literature with different biomasses and microorganisms in the removal of the same metal such as: the fungi *Paecilomyces* sp., *Penicillium* sp. and A. *niger* [33], for different water samples with XAD-4 resin modified with *A. kestanbolienis* [36], the adsorption of cobalt ions was evaluated using sediment samples from water bodies [48], the restoration of different soils and water by different biosorbents [49], the accumulation of copper and cobalt by *E. crassipes* [17], the bioremediation of waters contaminated with different heavy metals with the biomasses of the green bitter orange peel (*Citrus aurantium*), kumquat (*Citrus japonica*) and fine palomino grape seed (*Vitis vinifera* "Palomino") [50].

An in vivo experiment was also carried out with water lily plants, incubating them in presence and absence from water contaminated with the metal, from a lagoon of industrial effluents, finding that at 4 weeks of incubation, 17.3% of the metal under study is removed. In this regard, it has been reported that the water hyacinth can survive in presence of 60 $\mu$M of copper and cobalt during 60 days of incubation [17], the survival of water hyacinth in presence of a mixture of heavy metal concentrations of up to 3 mg/L, while at high concentrations of some metals (100 mg Cd/L), they caused a rapid discoloration of the plants [51], the removal of 0.02 mg/L of cadmium (II) from acid mining drainage with *E. crassipes*, for 14 days [52], a removal efficiency of 83.57% from simulated water with 2 mg/L of cadmium (II) using *E. crassipes* during 11 days [20], the removal of arsenic, cadmium, and Cupper from artificial wetlands for 60 days at a pH of 6.5, with different plants [22], the removal of 71% of 0.325 mg/L of mercury of wastewater from the mining industry for 7 months [25], and the phytoremediation of Nickel and Lead from soil and waters contaminated, using the same plant (water lily) [53].

## 5. Conclusions

In this study, cobalt (II) removal by the water lily biomass was analyzed. The performance of the biosorbent was examined as a function of the operating conditions, in particular: incubation time, pH, temperature, initial metal ion, and biomass concentration, as well as the phytoremediation of water contaminated in vivo and in vitro. The experimental evidence shows a strong effect of the experimental conditions. Maximum biosorption capacity values showed that the biosorbents used are very effective in the removal of cobalt (II) from aquatic systems in the conditions analyzed. Where the ease of production and economical parameters are concerned, it was observed that *E. crassipes* biomass, is a very promising biomaterial for removal or recovery of the heavy metal studied.

**Author Contributions:** Conceptualization, I.A.-R., A.R.-P. and N.C.P.-C.; methodology, I.A.-R., E.E.-D., A.R.-P. and J.F.C.-G.; software, J.F.C.-G.; validation, I.A.-R. and V.-M.M.-J.; formal analysis, I.A.-R. and A.R.-P.; investigation, I.A.-R., J.F.C.-G., A.R.-P., and N.C.P.-C.; resources, E.E.-D.; data curation, J.F.C.-G. and V.-M.M.-J.; writing—original draft preparation, I.A.-R., N.C.P.-C. and E.E.-D.; writing—review and editing, I.A.-R., N.C.P.-C., J.F.C.-G., and V.-M.M.-J.; visualization, A.R.-P.; supervision, I.A.-R. and E.E.-D.; project administration, I.A.-R. and J.F.C.-G.; funding acquisition, I.A.-R. All authors have read and agreed to the published version of the manuscript.

**Funding:** This research received no external funding.

**Institutional Review Board Statement:** Not applicable.

**Informed Consent Statement:** Not applicable.

**Data Availability Statement:** No new data were created or analyzed in this study. Data sharing is not applicable to this article.

**Acknowledgments:** This research work was carried out thanks in part to the support of the Directorate of the Faculty of Chemical Sciences of the Autonomous University of San Luis Potosi and the Directorate of the Center for Research and Extension of the Middle Zone, El Balandrán.

**Conflicts of Interest:** The authors declare no conflict of interest.

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
