# Peer review of "Removal of Cobalt (II) from Waters Contaminated by the Biomass of Eichhornia crassipes"

_water, doi:10.3390/w13131725_

Round 1
Reviewer 1 Report
The manuscript submitted by authors related to evaluate the adsorbent capacity of the biomass of the water lily (E. crassipes) in the removal of Cobalt (II) in aqueous solution is the good effort made by authors. But manuscript is not well presented and many concerns and suggestion are provided below for author’s attention.
Kindly seriously revised the script by responding all the point made in the script.
- In the abstract section author have mention “At the metal and biomass concentrations 24 analyzed, the natural biomass showed a good removal capacity”. Used scientific words rather than using good removal capacity.
- Authors have mention “In addition, this completely removal the metal in situ (100 mg/L in contaminated water, at 7 days of incubation, with 10 g of natural biomass in 100 mL), and in vivo, only eliminate 17.3% in 4 weeks. So, it can be used to remove it from industrial wastewater”. I have strong concern about the adsorption efficiency of this adsorbent. Low concentration of 100 ppm Co(II) takes so long days to remove. How this will be practically useful for real life industrial wastewater removal.
- Compare the adsorption capacity of present work with that of reported literatures.
- What is the motivation behind this work?
- Under introduction section, I can find 90% of information only related to Eichhornia crassipes. Kindly know the few point while writing the introduction. Include a brief literature review.Depending on the overall length of your paper, it will be necessary to include a review of the existing literature already published in the field. Use the literature to focus in on your contribution; and elaborate on the rationale of your paper.
- Provide the information about water pollution, different methods of water purification and highlighting the importance of adsorption process.
- Following below sentences with paraphrases should be added along with relevant references.Heavy metal ions and many other toxic elements originating from various industries are continuously contaminating water, soil, and air [].[Journal of Molecular Liquids 241, 1091–1113, 2017; Carbohydrate Polymers 134, 646–656, 2015; American Journal of Chemistry and Applications 3 (2), 8-19, 2016].
- Please write your text in good English (American or British usage is accepted, but not a mixture of these). English language manuscript require editing to eliminate possible grammatical or spelling errors and to conform to correct scientific English.
- Under introduction section add sentences “Various heavy metal ion removal techniques have been widely employed to date, for example, chemical precipitation (hydroxide precipitation, sulfide precipitation, and chelating precipitation), ion exchange, adsorption, membrane filtration (ultrafiltration, reverse osmosis, nanofiltration, and electrodialysis), coagulation and flocculation, electrochemical treatment, and so on. The adsorption process is usually favored to eliminate heavy metal ions because of its high performance, ease of handling, availability of various adsorbents, and cost-effectiveness[a-c]”.[(a)Carbohydrate polymers 230, 115597, 2020; (b)Materials Today Chemistry 18, 100376, 2020; (c)Photocatalysts in advanced oxidation processes for wastewater treatment, 167-196, 2020; (d)International Journal of Biological Macromolecules 143, 60-75, 2020].
- The U.S. Environmental Protection Agency(EPA) is responsible for regulating the levels of substances in the drinking water Provide the USEPA limit of Co in drinking water.
- Provide the impact of Co contaminated drinking water to human beings and ecosystem.
- Provide the google location coordinate map from where water lily plant was obtained for study.
- Results and discussion are very weak. Need more elaborated and discuss in more details.
- In page 3, section 3.1. Effect of incubation time an pH. Authors have mention “ Initially, the bioadsorption of 100 mg/L of Cobalt (II) was analyzed, at different incubation times at the following pH values: 4.0, 5.0, 6.0, 7.0, and 8.0 finding that at pH 5.0, 114 73.1% of the metal was removed at 28 hours”. But I cannot find any discussion. What basically authors wanted to interpreted with this finding. More clearly scientific discussion are needed.
- Similarly with other section 3.2. Effect of temperature, 3.3. Effect of the concentration of Cobalt (II), 3.4. Effect of initial biomass concentration.
- Provide the error bars in each graphs provided in the script.
- Please discuss how you calibrated each instrument used in your methods section and the accuracy and precision.
- What standards did you use to check your instrument?
- Provide the regeneration study of adsorbents.
- How the salts impact the adsorption capacity of metal.
- Provide with adsorption isotherms and adsorption kinetics models study.
- Provide the manuscript with schematic plausible mechanism for adsorption of Co(II) and discuss the mechanism in details.
- Provide with different speciation of Co at different pH.
Author Response
The manuscript submitted by authors related to evaluate the adsorbent capacity of the biomass of the water lily (E. crassipes) in the removal of Cobalt (II) in aqueous solution is the good effort made by authors. But manuscript is not well presented and many concerns and suggestion are provided below for author’s attention.
Kindly seriously revised the script by responding all the point made in the script.
- In the abstract section author have mention “At the metal and biomass concentrations 24 analyzed, the natural biomass showed a good removal capacity”. Used scientific words rather than using good removal capacity.
it was corrected in the text
- Authors have mention “In addition, this completely removal the metal in situ (100 mg/L in contaminated water, at 7 days of incubation, with 10 g of natural biomass in 100 mL), and in vivo, only eliminate 17.3% in 4 weeks. So, it can be used to remove it from industrial wastewater”. I have strong concern about the adsorption efficiency of this adsorbent. Low concentration of 100 ppm Co(II) takes so long days to remove. How this will be practically useful for real life industrial wastewater removal.
it was corrected in the text: We mentioned this property of dead biomass, since in vivo removal is very little, and as the reviewer suggests it is very slow.
- Compare the adsorption capacity of present work with that of reported literatures.
it was corrected in the text, but in the discussion the results obtained with respect to others in the literature are analyzed
- What is the motivation behind this work?
The use of different natural biomass has been reported to accumulate and remove different pollutants, among which are heavy metals, and such as water Lily, and it is considered one of the worst weeds in the world, causing a great amount of damage to aquatic ecosystems, we look for applications that benefit the environment, such as its ability to bio-absorb heavy metals, in addition to being easy to obtain in large quantities and it is very economical
- Under introduction section, I can find 90% of information only related to Eichhornia crassipes. Kindly know the few point while writing the introduction. Include a brief literature review.Depending on the overall length of your paper, it will be necessary to include a review of the existing literature already published in the field. Use the literature to focus in on your contribution; and elaborate on the rationale of your paper.
It was corrected in the text.
- Provide the information about water pollution, different methods of water purification and highlighting the importance of adsorption process.
It was corrected in the text.
- Following below sentences with paraphrases should be added along with relevant references.Heavy metal ions and many other toxic elements originating from various industries are continuously contaminating water, soil, and air [].[Journal of Molecular Liquids 241, 1091–1113, 2017; Carbohydrate Polymers 134, 646–656, 2015; American Journal of Chemistry and Applications 3 (2), 8-19, 2016].
It was incorporated in the text
- Please write your text in good English (American or British usage is accepted, but not a mixture of these). English language manuscript require editing to eliminate possible grammatical or spelling errors and to conform to correct scientific English.
It was corrected in the text.
- Under introduction section add sentences “Various heavy metal ion removal techniques have been widely employed to date, for example, chemical precipitation (hydroxide precipitation, sulfide precipitation, and chelating precipitation), ion exchange, adsorption, membrane filtration (ultrafiltration, reverse osmosis, nanofiltration, and electrodialysis), coagulation and flocculation, electrochemical treatment, and so on. The adsorption process is usually favored to eliminate heavy metal ions because of its high performance, ease of handling, availability of various adsorbents, and cost-effectiveness[a-c]”.[(a)Carbohydrate polymers 230, 115597, 2020; (b)Materials Today Chemistry 18, 100376, 2020; (c)Photocatalysts in advanced oxidation processes for wastewater treatment, 167-196, 2020; (d)International Journal of Biological Macromolecules 143, 60-75, 2020].
It was incorporated in the text
- The U.S. Environmental Protection Agency (EPA) is responsible for regulating the levels of substances in the drinking water Provide the USEPA limit of Co in drinking water.
It was incorporated in the text in discussion
- Provide the impact of Co contaminated drinking water to human beings and ecosystem.
It was incorporated in the text in discussion
- Provide the google location coordinate map from where water lily plant was obtained for study.
the location in material and methods was added
- Results and discussion are very weak. Need more elaborated and discuss in more details.
It was corrected in the text.
- In page 3, section 3.1. Effect of incubation time an pH. Authors have mention “ Initially, the bioadsorption of 100 mg/L of Cobalt (II) was analyzed, at different incubation times at the following pH values: 4.0, 5.0, 6.0, 7.0, and 8.0 finding that at pH 5.0, 114 73.1% of the metal was removed at 28 hours”. But I cannot find any discussion. What basically authors wanted to interpreted with this finding. More clearly scientific discussion are needed.
See discussion
- Similarly with other section 3.2. Effect of temperature, 3.3. Effect of the concentration of Cobalt (II), 3.4. Effect of initial biomass concentration
See discussion
.
- Provide the error bars in each graphs provided in the script.
It was corrected in the text.
- Please discuss how you calibrated each instrument used in your methods section and the accuracy and precision.
It was corrected in the text.
- What standards did you use to check your instrument?
was corrected in the text.
- Provide the regeneration study of adsorbents
it is not understood what this suggestion refers to
- How the salts impact the adsorption capacity of metal.
as the sample was washed with edta and tridesionized water, contaminants were removed so that they did not interfere with the removal studies, it has also been reported that the salts do not significantly interfere with removal
- Provide with adsorption isotherms and adsorption kinetics models study.
we don't think it's necessary
- Provide the manuscript with schematic plausible mechanism for adsorption of Co(II) and discuss the mechanism in details.
The mechanism by which cadmium is removed by the water lily biomass has not been described, and with the data obtained, we believe that we cannot propose a mechanism
- Provide with different speciation of Co at different pH.
there's no need,
Submission Date
19 May 2021
Date of this review
20 May 2021 10:01:43
Final del formulario
©

Reviewer 2 Report
This manuscript studied Co(II) removal from aqueous phase by lily biomass and fresh lily. This study matches the scope of Water. However, the manuscript is not well-written and some complementary work is required. I would recommend the publication of manuscript after a major revision. 1 Although the English-writing of this manuscript is generally standard, the improvement of English is still required. For example in abstract, “So the objective of this work was to determine the removal of Cobalt (II) in aqueous solution by the biomass of the aquatic lily or water hyacinth (Eichhornia crassipes) which, is one of the main weeds 18 present in fresh water, due to its rapid reproduction, growth and high competitiveness, by the colorimetric method of the Methyl isobutyl ketone” 2 How authors define “bio-adsorption”? If the adsorption of Co(II) on lily is still due to the complexation of Co(II) with surface functional groups , I do not think it is so-called bio-adsorption. 3 Authors did not explain clearly why treated fresh lily with many processes. I think similar treatment methods are available in the literature. Authors should take into account these methods and explain how the condition parameters (like time, temperature and so on) were selected. 4 Authors should also test and show the repeatability of lily bio-mass, i.e., whether lily bio-mass in different batches have similar composition and performance. 5 There is no characterization on lily bio-mass. I think at least FT-IR, BET and XRD are required. This is also important to support that lily bio-mass is a promising adsorbents.Author Response
This manuscript studied Co(II) removal from aqueous phase by lily biomass and fresh lily. This study matches the scope of Water. However, the manuscript is not well-written and some complementary work is required. I would recommend the publication of manuscript after a major revision.
1 Although the English-writing of this manuscript is generally standard, the improvement of English is still required. For example in abstract, “So the objective of this work was to determine the removal of Cobalt (II) in aqueous solution by the biomass of the aquatic lily or water hyacinth (Eichhornia crassipes) which, is one of the main weeds 18 present in fresh water, due to its rapid reproduction, growth and high competitiveness, by the colorimetric method of the Methyl isobutyl ketone”
it was corrected in the tex
2 How authors define “bio-adsorption”? If the adsorption of Co(II) on lily is still due to the complexation of Co(II) with surface functional groups , I do not think it is so-called bio-adsorption.
it was corrected in the tex
3 Authors did not explain clearly why treated fresh lily with many processes. I think similar treatment methods are available in the literature. Authors should take into account these methods and explain how the condition parameters (like time, temperature and so on) were selected.
it was corrected in the tex
4 Authors should also test and show the repeatability of lily bio-mass, i.e., whether lily biomass in different batches have similar composition and performance.
This suggestion is very interesting but we think that it is not relevant for this study, since it was found that this biomass efficiently removes the metal, which is the objective of the work, and we believe that if the biomass is taken in other seasons of the year, the removal percentage of metal removal, and the composition for this job is not very important
5 There is no characterization on lily bio-mass. I think at least FT-IR, BET and XRD are required. This is also important to support that lily bio-mass is a promising adsorbents.
It would be fabulous, but we don't have the methodology or the resources, because if we send them done they are very expensive

Reviewer 3 Report
The work of Ismael et al. proposes an determination of Cobalt via its bioaccumulation in Eichhornia crassipes.
The title is concise and the abstract states clearly the work performed, as well as the results.
Similar studies took place more than a decade ago on another macrophyte, Lemna minor, therefore there is not much novelty involved from this point.
Cobalt usually occurs in the environment in association with other metals such as copper, arsenic and nickel. In one oral study, no developmental effects on human fetuses were observed following treatment of pregnant women with cobalt chloride [1]. Furthermore, no correlation was found between cancer mortality and the level of cobalt in the water [1].
However, heavy metals above a safe limit pose a threat to human health and to the aquatic environment as in long therm exposure.
The work of Ismael et al. brings a contribution to the current literature from the point of view that the mechanism of bioacumulation in E. crassipes was not fully studied so far.
The work is well written and the clear and simplistic approach is to be appreciated.
The introduction is well written and well cited.
Materials and methods
Line 101: to what pH was it adjusted?
Line 104-106: " Cobalt (II) concentration in aqueous solution was determined by UV spectroscopy in a double-beam UV-Visible spectrophotometer Shimadzu UV-2101PC, using Methyliso-butylketone" Describe the method used from the cited " Colorimetric Determination of Elements"1964, as only the title seem to be in English. The purpose of a paper is to be reproducible!
Line 158: A simple chlorophyll content spectrophotometric determination should have been performed, I suggest to perform it if the material is still available.
The discussion and conclusions are well written and stated.
We need to appreciate scientist that have to use limited resources in order to perform their work, it is a burden we all had in the beginning of our careers. Although the paper is interesting enough, the above mentioned need the authors attention.
I propose a minor revision especially underlining the fact that Line 104-106 has to be detailed with the method used, step by step.
References
1. Agency for Toxic Substances and Disease Registry (ATSDR). Toxicological Profile for Cobalt. Public HealthService, U.S. Department of Health and Human Services, Atlanta, GA. 1992
Author Response
The work of Ismael et al. proposes an determination of Cobalt via its bioaccumulation in Eichhornia crassipes.
The title is concise and the abstract states clearly the work performed, as well as the results.
Similar studies took place more than a decade ago on another macrophyte, Lemna minor, therefore there is not much novelty involved from this point.
Cobalt usually occurs in the environment in association with other metals such as copper, arsenic and nickel. In one oral study, no developmental effects on human fetuses were observed following treatment of pregnant women with cobalt chloride [1]. Furthermore, no correlation was found between cancer mortality and the level of cobalt in the water [1].
It was corrected in the text
However, heavy metals above a safe limit pose a threat to human health and to the aquatic environment as in long therm exposure.
The work of Ismael et al. brings a contribution to the current literature from the point of view that the mechanism of bioacumulation in E. crassipes was not fully studied so far.
The work is well written and the clear and simplistic approach is to be appreciated.
The introduction is well written and well cited.
Materials and methods
Line 101: to what pH was it adjusted?
It was corrected in the text
Line 104-106: " Cobalt (II) concentration in aqueous solution was determined by UV spectroscopy in a double-beam UV-Visible spectrophotometer Shimadzu UV-2101PC, using Methyliso-butylketone" Describe the method used from the cited " Colorimetric Determination of Elements"1964, as only the title seem to be in English. The purpose of a paper is to be reproducible!
It was corrected in the text
Line 158: A simple chlorophyll content spectrophotometric determination should have been performed, I suggest to perform it if the material is still available.
The material is no longer available, as it was treated and processed to obtain biomass
The discussion and conclusions are well written and stated.
We need to appreciate scientist that have to use limited resources in order to perform their work, it is a burden we all had in the beginning of our careers. Although the paper is interesting enough, the above mentioned need the authors attention.
I propose a minor revision especially underlining the fact that Line 104-106 has to be detailed with the method used, step by step.
It was corrected in the text
References
- Agency for Toxic Substances and Disease Registry (ATSDR). Toxicological Profile for Cobalt. Public HealthService, U.S. Department of Health and Human Services, Atlanta, GA. 1992
Submission Date
19 May 2021
Date of this review
24 May 2021 20:59:36

Round 2
Reviewer 1 Report
The authors have satisfactory revised the manuscript.
Reviewer 2 Report
The revised manuscript is ready for publication.